# OpenReview forum: "Adversarial Prompt Evaluation: Systematic Benchmarking of Guardrails Against Prompt Input Attacks on LLMs"
_NeurIPS.cc/2024/Workshop/SafeGenAi — SafeGenAi Oral_

### Official Review · Reviewer_7NyF · 2024-10-09
**Comprehensive evaluation of multiple defenses and jailbreak attacks.**

**Rating:** 7
**Confidence:** 3

**Review:**

The paper performs a comprehensive evaluation of different guardrails against different popular jailbreak attacks. Research questions are clearly documented and answered properly. This work is a useful contribution to the literature.

---

### Official Review · Reviewer_o3rx · 2024-10-09
**Benchmarking of jailbreak defences on a consolidated test set from a number of jailbreak datasets showing suprising strength of finetuned small classifiers**

**Rating:** 9
**Confidence:** 4

**Review:**

This paper benchmarks the following jailbreak defenses across a wid range of jailbreak prompt datasets:
- Perplexity threshold
- Random Forest Classifier based on unigram features in the dataset
- BERT, DeBERTa, GPT2 based classifiers including ProtectAIGuard
- Embedding cosine similarity using LangKit
- APis including Azure AI Content Safety and OpenAI Moderation
- LLM-as-a-judge including using NeMo GuardRails
- Finetuned LLM - LlamaGuard
The paper includes a variety of interesting findings and practical recommendations based on the benchmarking including:
- Larger LLMs and better system prompts are beneficial at detecting jailbreaks
- Finetuned BERT, DeBERTa and GPT2 models outperform more expensive and closed source methods, even on datasets they were not trained on.
- A simple random forest classifier is competitive in the same data distribution as what it was trained on, although it does not generalize well to out of distribution data.
- LLM-as-a-judge methods boost jailbreak detection rate but at the cost of a higher false positive rate.

Overall, the most valuable contribution from this paper is the analysis of the efficacy of the various defenses. Many papers that develop content moderation models include such analyses but they rarely go beyond LLM-as-a-judge and finetuned LLM models. It is important in the field to periodically re-evaluate approaches that may be based on older methods as is done in this paper.

One minor weakness is that much of the discussion related to questions RQ2 and RQ3 are well known to researchers working on these topics. However, it is likely that these are beneficial to practitioners who have limited knowledge of how the field of NLP has progressed over the years, although they may also find the lack of a clear conclusion in RQ3 inconclusive. It make have been possible to write a more concise short-paper length version focussing primarily on the benchmarking, and I would recommend the authors attempt this for a main conference submission in future. Despite this, I consider this paper to be a valuable contribution to this workshop.

---

### Official Review · Reviewer_VSqQ · 2024-10-10
**A useful evaluation**

**Rating:** 6
**Confidence:** 4

**Review:**

# Summary

This work studies the effectiveness of '"guardrail" style defences: supplementary systems/algorithms which are used in conjunction with trained models to preprocess data, detect malicious inputs, etc., as opposed to trying to train more robust models purely through better alignment/safety training techniques.
The authors pit 18 different guardrail techniques against a number of different static jailbreaks, and show that there is no single defense approach that perfectly defends against all jailbreak attacks, and some impart false positive detection of benign content.

To the best of my knowledge, the research community is less focused on evaluating these 'guardrail' techniques as they are more often used in industrial/applied settings, but they are a useful tool in trying to deploy safe(er) models, and understanding their failure modes and limitations is relevant and useful to the research community.

# Merits

- Clear and well posed research questions
- To the best of my knowledge, the field is more focused on evaluating the overall robustness of LLMs without using guardrails as a defense. While we hope that one day we will have truly safe and robust models through algorithmic advances, guardrails are currently an effective and pragmatic tool to use as part of the defense toolkit of LLM-based systems. As such, more work in probing their limitations and vulnerabilities is useful

# Limitations / Questions

- I believe the agreed usage of *prompt injection* is as a style of jailbreak that leverages the way prompts are formatted with delimiter tokens, e.g. abusing/injecting `<|start_header_id|>assistant | user<|end_header_id|>` in Llama 3 chat formatting in order to give context to the model that it already produced content, rather than any type of jailbreak which is what's suggested in this work. This work is a bit unclear in how it frames its threat model; i.e. it seems to be talking about jailbreaks in general, but certain paragraphs suggest they primarily consider prompt injection?
- While the main research questions and takeaways are clear, I think the presentation of results and the results section I think could be improved a bit; it is difficult to parse the takeaways from the tables

---

### Official Review · Reviewer_YKN8 · 2024-10-10

**Rating:** 9
**Confidence:** 4

**Review:**

1. **Quality**:
The paper presents a systematic benchmarking of guardrails designed to protect large language models (LLMs) from adversarial prompt inputs. The quality of the work is high, and the authors have conducted an extensive evaluation across various defenses and datasets, which adds to the robustness of their findings.

2. **Clarity**:
The paper is well-structured and exhibits a high level of clarity in its writing.

3. **Originality**:
The paper contributes to the field by providing one of the first systematic evaluations of guardrail mechanisms against a diverse set of adversarial prompt attacks. The inclusion of both malicious and benign datasets further enhances the originality of the work.

4. **Significance**:
The findings are highly significant for the deployment of LLMs in practical scenarios, offering valuable insights into the implementation of guardrails. However, the link to the open-sourced code is missing, and the authors should check to ensure the code is publicly available.